# Multifrequency radar observations of marine clouds during the EPCAPE campaign

Juan M. Socuellamos[1], Raquel Rodriguez Monje[1], Matthew D. Lebsock[1], Ken B. Cooper[1], Robert M. Beauchamp[1] and Arturo Umeyama[1]

[1]Jet Propulsion Laboratory, California Institute of Technology, Pasadena, CA 91109, USA

*Correspondence to*: Raquel Rodriguez Monje (raquel.rodriguez.monje@jpl.nasa.gov)

**Abstract.** The Eastern Pacific Cloud Aerosol Precipitation Experiment (EPCAPE) was a year-round campaign conducted by the US Department of Energy at the Scripps Oceanographic Institute in La Jolla, CA, USA, with a focus on characterizing atmospheric processes at a coastal location. The ground-based prototype of a new Ka, W and G-band (35.75, 94.88 and 238.8 GHz) profiling atmospheric radar, named CloudCube, and developed at the Jet Propulsion Laboratory, took part in the experiment during six weeks in March and April, 2023. This article describes the unique data sets that were obtained during the field campaign from a variety of marine clouds and light precipitation. These are, to the best of the authors' knowledge, the first observations of atmospheric clouds using simultaneous multifrequency measurements including 238.8 GHz. These data sets therefore provide an exceptional opportunity to study and analyze hydrometeors with diameters in the millimeter and submillimeter size range, that can be used to better understand cloud and precipitation structure, formation, and evolution. The data sets referenced in this article are intended to provide a complete, extensive, and high-quality collection of G-band data, in the form of Doppler spectra and Doppler moments. In addition, Ka and W-band reflectivity and Ka, W and G-band reflectivity ratio profiles are included for several cases of interest on six different days. The data sets can be found at https://doi.org/10.5281/zenodo.10076227 (Socuellamos et al., 2024).

## 1 Introduction

Coastal environments adjacent to cities and industry offer unique opportunities to study and analyze the effects of aerosols on cloud and precipitation formation and evolution (Sanchez et al., 2016). Moreover, the seasonal temperature gradient between the sea/ocean mass and the lower atmosphere, together with the coastal orography, commonly generates a thin low-altitude marine cloud cover containing generally small hydrometeors that can conveniently be used to study cloud formation and evolution, interaction between hydrometeors and aerosols, and surface-atmosphere radiation exchange, with the goal of improving weather models and prediction (Petters et al., 2006; Lin et al., 2009). This is the focus of the Eastern Pacific Cloud Aerosol Precipitation Experiment (EPCAPE; Russell et al., 2021), a field campaign promoted by the US Department of Energy, that hosted different types of instruments to be deployed at different locations at the Southern California coastal line.

The response of clouds and cloud processes to warming are the main physical source of uncertainty in climate prediction (Zelinka et al., 2017). Furthermore, model representations of the radiative forcing of clouds due to their interaction

with aerosols vary by a factor of two (Boucher et al., 2013). In addition, the large-scale effects are difficult to characterize because they result from small-scale processes (Baker and Peter, 2008). For both the cloud-climate feedback and aerosol-cloud interactions, the droplet collection process that governs the initiation of precipitation has been implicated as an important source of uncertainty (Jing and Suzuki, 2018; Mülmenstädt et al., 2021). In particular, drops with diameters in the

submillimeter range are the embryonic precipitation drops for which there is currently a significant observational gap. This motivates the use of millimeter and submillimeter-wave remote sensing instrumentation, capable of profiling inside clouds and precipitation with fine vertical resolution, to properly analyze the microphysics and dynamics of these atmospheric processes. Radars are a particularly suitable fit for these kinds of measurements as they can generally penetrate longer distances than laser-based instruments and profile inside clouds and precipitation with finer resolution than state-of-the-art radiometers.

Hydrometeors possess variable and identifiable absorption and scattering properties that cause them to interact differently with a radar's transmitted signal depending on its frequency (Leinonen et al., 2015). The use of a millimeter-wave multifrequency radar, with simultaneous measurements of the same atmospheric structure at different frequency bands including the G-band, can be used to characterize particle size distributions with drop sizes in the submillimeter range, and to detect small amounts of liquid water content, revealing new valuable information about cloud and precipitation behavior

(Battaglia et al., 2014). In addition, the combination of G-band Doppler radar with lower frequency channels offers significant benefits for quantifying the properties of ice-phase hydrometeors. As suggested by Battaglia et al. (2014), using dual-frequency reflectivity ratios from three different channels including G-band has the potential to identify snow crystals habit, while Hogan et al. (2000) point out the utility of the G-band dual-frequency ratio for sizing cirrus crystals. With the burgeoning availability of multifrequency radar observations including G-band (Lamer et al., 2021, Courtier et al., 2022), the coming years offer a

tremendous opportunity to validate these theorized remote sensing capabilities.

        CloudCube, a new multifrequency (Ka, W and G-band) radar developed at the Jet Propulsion Laboratory (JPL) under the National Aeronautics and Space Administration Earth Science Technology Office (NASA-ESTO) Instrument Incubator Program (IIP), aims to tackle some of the most relevant Earth Science questions by exploiting the differential hydrometeor-signal interaction to provide novel insight into clouds and precipitation microphysics and dynamics. CloudCube measures

vertical profiles of reflectivity at each frequency band and Doppler spectra at G-band, enabling a uniquely detailed analysis of the smallest hydrometeors.

        After recently completing the development of the three CloudCube prototype radars (35.75, 94.88 and 238.8 GHz), built for ground and airborne validation, we joined the EPCAPE field campaign during six weeks in the months of March and April, 2023. While Ka-band and W-band observations are extensively available in the literature, the data sets provided and

discussed in this article contain, to the best of the authors' knowledge, the first measurements of clouds and precipitation above 200 GHz, and the first simultaneous multifrequency measurements that include 238.8 GHz. Moreover, CloudCube provides enhanced sensitivity and vertical resolution compared to previous G-band radars (Courtier et al., 2022), making possible to extend the hydrometeor study to smaller particles never analyzed before. The G-band data sets contain the Doppler spectra and Doppler moments from diverse cloud structures and light precipitation. In addition, Ka and W-band reflectivity and Ka,

W and G-band reflectivity ratio profiles have also been included for several cases of interest on six different days. This article begins with a brief description of the three CloudCube modules and the participation in the field campaign, to later explain how the raw data from the observations have been processed and made available to the scientific community.

## 2 Instrument and observations

### 2.1. CloudCube instrument

CloudCube's radar architecture relies on all-solid-state technology, and uses the offset I/Q (in-phase and quadrature) modulation technique with pulse compression. This design achieves high radar sensitivity, while significantly reducing the overall size, weight, and power consumption (SWaP) of the instrument. This approach follows that of RainCube, a Ka-band spaceborne precipitation radar in a CubeSat developed previously by JPL (Beauchamp et al., 2017; Peral et al., 2018a; Peral et al., 2018b). The G-band radar, in a prototype stage, was operated on frequency modulated continuous wave (FMCW) mode

during this deployment to eliminate the blind range and improve the sensitivity, and included Doppler capability to complement the multifrequency measurements. CloudCube's W- and G-band modules are also built to validate, for the first time, the I/Q direct up/down-conversion approach at these high frequencies, a major step to achieve a compact radar architecture and to enable the subsequent design of flight-ready instruments compatible with low-cost satellite platforms to facilitate multi-instrument or constellation missions (Tanelli et al., 2018; Stephens et al., 2020).

The three CloudCube frequency channels deployed in EPCAPE are built from discrete, commercially available or JPL-designed RF components and assembled into three separate rack-mounted chassis. Each module contains two main subsystems: the radar transceiver to generate the millimeter-wave signal and detect the target echo, and the digital processor where the chirped waveform is created and the received echo is acquired and processed. The baseband signal is directly upconverted to RF without any intermediate stages reducing the number of discrete RF components and the overall size of the

radar. The CloudCube modules that have been operated during the EPCAPE field campaign are shown in Fig. 1, and the radar parameters used to record the data presented in this article are summarized in Table 1. The information in Table 1 has been included as global attributes in the provided data sets. Different pulse widths and pulse repetition intervals were used to characterize the radars performance.

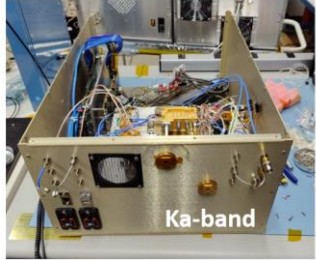 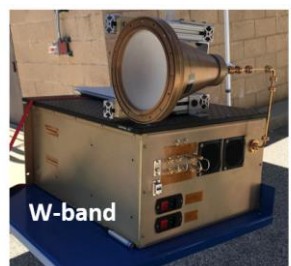 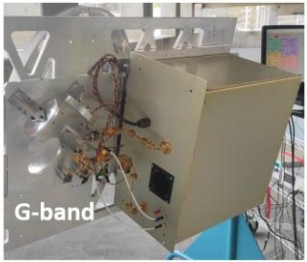

**Figure 1: Pictures of the CloudCube rack-mounted prototype modules operated during the EPCAPE deployment. From left to right: the Ka-band, W-band and G-band CloudCube radar channels.**

**Table 1: Radar parameters of the three frequency channels of CloudCube's ground-based prototypes during the EPCAPE field campaign.**

|  | Ka-band | W-band | G-band |
|---|---|---|---|
| **Frequency (GHz)** | 35.75 | 94.88 | 238.8 |
| **Transmission type** | Pulsed | Pulsed | FMCW |
| **Pulse width (μs)** | 1, 5, 150 | 1, 5, 10, 20 | 40, 60 |
| **Pulse repetition interval (ms)** | 0.35, 0.5, 1, 2 | 0.35, 0.5, 1, 2 | 0.042, 0.084 |
| **Chirp bandwidth (MHz)** | 0, 2 | 0, 2 | 15 |
| **Peak transmit power (W)** | 10 | 10 | 0.08, 0.24 |
| **Antenna diameter (cm)** | 30 | 30 | 60 |
| **Sensitivity at 1 km (dBZ)** | -10 | -15 | -40 |
| **Unambiguous range (km)** | 52.5, 75, 150, 300 | 52.5, 75, 150, 300 | 6.3, 12.6 |
| **Range resolution (m)** | 75, 150 | 75, 150 | 10 |
| **Unambiguous velocity (ms$^{-1}$)** | - | - | ±7.5, ±3.75 |
| **Velocity resolution (ms$^{-1}$)** | - | - | 0.06, 0.03 |


## 2.2. EPCAPE deployment

The EPCAPE campaign was conducted at the Scripps Oceanographic Institute in La Jolla, CA, USA, where the Ellen Browning Scripps Memorial Pier served as the main site for operations (see Fig. 2a). Prior to the beginning of the deployment, we installed CloudCube in a trailer with apertures on the roof through which the radars were looking upwards to perform the

observations. These apertures were not complemented with the installation of radomes, so the observations were limited to clouds and drizzle to avoid instrument damage from rain. Since the radars were pointing zenith in this configuration, we have used range and height interchangeably in this manuscript to describe the targets' distance to the radars. Along with CloudCube, the JPL-developed 170-GHz Vapor In-cloud Profiling Radar (VIPR, Cooper et al., 2021), was deployed to profile water vapor content inside clouds.

CloudCube's Ka-band channel, which is a built-to-print replica of RainCube's spaceborne hardware, was configured as a bistatic instrument for this deployment. This configuration was adopted to circumvent the significant blind range inherent in the spaceborne hardware's legacy (Peral et al., 2018b). In contrast, we retained the monostatic configuration of the W-band radar and made use of short pulses where possible to minimize the blind range. The G-band module uses a quasi-optical duplexing system with a large primary reflector. The quasi-optical duplexing system provides excellent isolation between the

transmit and receive ports (Cooper et al, 2012) allowing the operation of the instrument in frequency-modulated continuous-wave (FMCW) mode with no blind range. Fig. 2b shows the different CloudCube modules and VIPR as installed in the trailer during observations.

Along with the JPL trailer, Fig. 2a also shows the U.S. Department of Energy (DoE) Atmospheric Radiation Measurement (ARM) user facility, that operated multiple instruments, including radiometers, lidars and additional radars, in parallel to CloudCube.

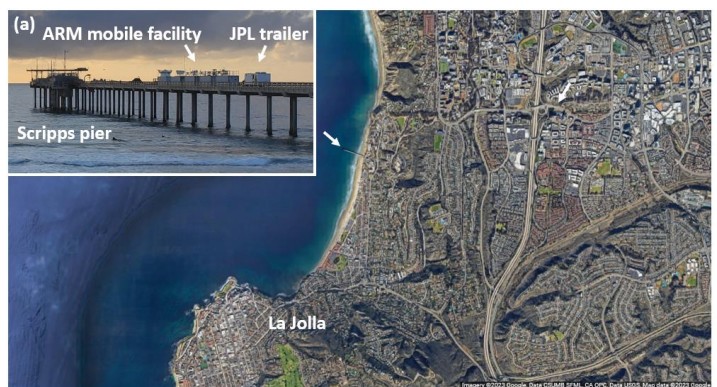 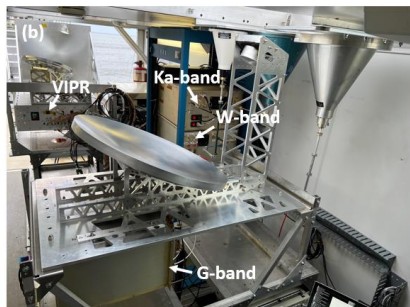

**Figure 2: Location and deployment of the CloudCube instrument during the EPCAPE field campaign. (a) The EPCAPE experiment is conducted at the Scripps Oceanographic institute in La Jolla, CA, USA. (b) Picture inside the trailer where VIPR and CloudCube were installed and operated from.**

### 2.3. Data selection

CloudCube was operated on-site on weekdays for approximately 12 hours (from 6 am to 6 pm Pacific Time) for six weeks starting on March 23 and ending on April 27. However, the instrument was operated only when cloud targets were present. Therefore, the data sets are provided on a target-detection basis, and not as continuous 12-hour recordings.

In addition, other factors limit the data availability:

- During the first week of operation, March 23 and March 24, only the W-band and G-band modules of CloudCube were installed. The Ka-band radar was added on the next week, March 30, and data with three-frequency measurements are only available from that day onward.

- We set the G-band instrument parameters as described in Table 1 finding a good compromise between the unambiguous range and Doppler velocity. For the majority of the cases, we operated with 6.3 km and 7.5 ms$^{-1}$ unambiguous range and velocity, respectively. While we did not observe hydrometeor velocities higher than the maximum unambiguous velocity, we did have a few days with high-level clouds above the maximum unambiguous range that appeared as low/mid-level clouds in the folded (aliased) range-Doppler spectrum. We have addressed this issue in the provided data sets by unfolding the echo signals to correctly represent the target altitudes (see Sect. 3.3). However, when low-level and high-level clouds were present at the same period and coincident in the folded spectrum, they appeared as overlapped echo signals, preventing the differentiation of the target features and altitude. These data, obtained on April 12, have been discarded.

- Close-range marine stratocumulus clouds, fog and drizzle were a common occurrence during the period that CloudCube operated and we have provided extensive data including those cloud types. However, given the monostatic and pulsed-mode configuration of the W-band radar, and the use of a switch system to avoid damage to the receiver components that carries additional timing, W-band data are typically not available for approximately the first 500 m.

The data availability is summarized in Fig. 3, sorted by the days of observations and the different atmospheric conditions. From March 23 to March 30, low-level (altitudes lower than 2 km) and mid-level (altitudes between 2 km and 7 km) stratocumulus and cumulonimbus clouds with sporadic periods of precipitation were dominant. On March 31 and April 1, mid/high-level cirrus clouds were observed. Close-range thin marine and high-level cirrus clouds were present and coincident on April 11 and April 12. Finally, on April 3 and from April 13 to the end of CloudCube's participation in the experiment, low-level marine stratocumulus clouds were predominant. Missing days in Fig. 3 are due to clear-sky conditions during which we did not operate the instrument.

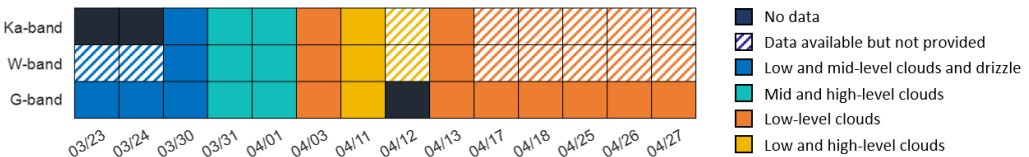

**Figure 3: CloudCube's data availability and classification during the participation in the EPCAPE field campaign in March and April 2023. The days marked with a dashed pattern refer to days where data are available but have not been provided to avoid repetition of similar observations and maintain a manageable number of files and data package sizes. These data can be provided upon request to the corresponding author.**

An example of multifrequency reflectivities that can be found in the data provided with this article is plotted in Fig. 4. The combination of simultaneous observations at three greatly spaced frequency bands can reveal distinct cloud and precipitation features to further enhance the microphysical analysis. The process to obtain the calibrated data in Fig. 4, as well as dual-frequency ratios, and G-band Doppler spectra and moments, is described in Sect. 3.

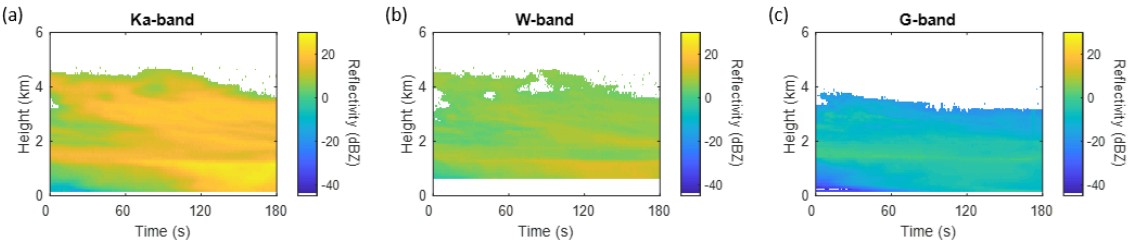

**Figure 4: Example of CloudCube data on March 30, starting time 17:12:52 UTC, showing calibrated reflectivity at Ka (a), W (b) and G-band (c). The W-band plot (b) shows no data for approximately the first 500 m, corresponding to the blind range of the radar.**

## 3. Data processing

### 3.1. Overview

The final data products that are described in this article have gone through several steps to provide calibrated reflectivity and to enhance the overall quality of the data sets. A flowchart of the process illustrating the different steps followed to obtain the final data products is shown in Fig. 5. Initially, we applied a data quality control process that included selecting relevant observations, removing noise and artifacts, and, in the case of the G-band data, unfolding the G-band Doppler spectra where possible (step 1). We then applied a calibration factor to the G-band Doppler spectra data (2), previously obtained from

an absolute calibration of the radar, to obtain calibrated spectral reflectivity and form the first data product (3). Subsequently, we calculated the G-band Doppler moments (4), which constitute the second data product discussed in this article. Finally, we utilized the G-band Doppler moments to identify optimal atmospheric formations to cross-calibrate the W-band and Ka-band raw data using the G-band absolute calibration as reference (5). After spatiotemporally matching the calibrated data and subtracting the gaseous attenuation at the three different frequency bands, we produced the third and final data product which

includes multifrequency reflectivity and dual-frequency reflectivity ratios (6). The different steps in CloudCube's data processing are described in more detail in the following subsections.

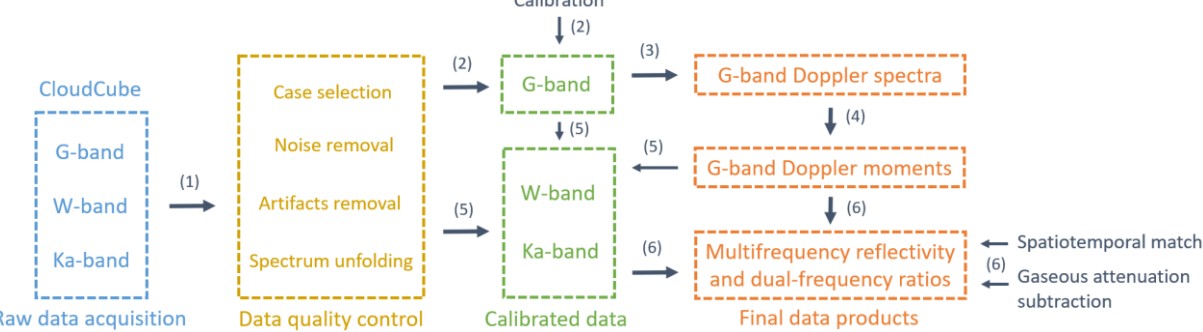

**Figure 5: CloudCube's data processing flowchart. A data quality control and calibration process were applied to the raw data to produce three separate data sets: G-band Doppler spectra, G-band Doppler moments, and Ka, W, and G-band multifrequency**
**reflectivity and dual-frequency ratios.**

### 3.2. G-band calibration

      One of the main goals of a multifrequency instrument such as CloudCube is to be able to compare the differential scattering signatures of hydrometeors that can be exploited to obtain new insight into cloud microphysical processes. The

comparison of the differential signals, and the information obtained from it, can only be trusted when the instruments are properly calibrated and the quantitative data are reliable.

      In preparation for the participation in the field campaign, we calibrated the G-band radar carefully pointing the instrument towards a metal sphere with radius $r_s = 10$ cm at a distance of approximately $d_s = 600$ m. The echo return, $P_s$, from a target with a well-known cross-section, can be compared to a theoretical model to calculate a calibration factor to be later

applied to measurements of atmospheric targets with unknown cross-sections (Atlas and Mossop, 1960). Then, the arbitrary amplitude levels displayed on our digital processor can be converted to observed reflectivity values. For that purpose, we followed the expression in Roy et al. (2020)

$$C_G = \frac{\lambda_G^4 \sigma_s e^{-2\beta_G}}{\pi^5 |K_G|^2 \Omega_G r_s^4 \Delta r_G P_s},$$
(1)

where $\lambda_G$ is the wavelength of the transmitted signal, $\sigma_s$ is the cross-section of the spherical target, $\Omega_G$ is the antenna solid angle

and $\Delta r_s$ is the range resolution. $\beta_G$ and $K_G$ are the optical depth and the dielectric ratio, respectively, and are weather-dependent variables that we calculated using ITU (2013) and Elton (2016), respectively. Uncertainties in the determination of the calibration factor may arise from an inaccurate knowledge of the radar parameters and weather conditions needed as input values in Eq. (1), or from an imperfect alignment of the calibration sphere to the radar beam center. While these uncertainties are difficult to quantify precisely, Roy et al. (2020) estimate that they may lead to an error of around 1 dB in the final calibrated

reflectivity values.

        The calibration was performed using a transmitter source of $P_{t-} = 80$ mW, the same source that we used on March 23 and March 24 during the field campaign. From March 30 onward, we replaced the transmitter source, increasing the transmit power to $P_{t+} = 240$ mW. If we had used this higher-power source during calibration, the echo power would have been increased by the same amount, i.e. we would have obtained an echo amplitude three times higher compared to what we obtained with

the lower-power source. We then corrected the calibration factor to account for that higher transmitted power as

$$C_{G+} = \frac{P_{t-}}{P_{t+}} C_{G-},$$
(2)

and applied this new factor to the data sets where the higher-power source was employed.

### 3.3. G-band Doppler spectra and moments

The G-band radar, as an instrument with Doppler capability, provides information about observations in the form of velocity-range spectra. An example of real-time data, as obtained during operation after averaging 256 collected pulses, is shown in Fig. 6a, where the negative Doppler velocity corresponds to targets moving toward the radar, i.e. falling hydrometeors. Using the calibration factor obtained in Sect. 3.2, the amplitude values shown in Fig. 6a can be translated into observed spectral reflectivity data following the expression (Doviak and Zrnic, 1993)

$$Z_{G,s}(v) = C_G r^2 P_G(v),$$
(3)

with $r$ being the range at which the target is detected, and $P_G(v)$ being the echo amplitude in the velocity-range spectrum.

        Prior to that, the echo signal represented as amplitude in arbitrary dB units in Fig. 6a, was processed to subtract the noise floor and obtain a cleaner spectrum. In order to find the noise values to be subtracted from our measurements, we produced histograms representing the noise and signal distribution with height as shown in Fig. 6b. We have taken advantage

of the full Doppler velocity span (see Fig. 6a) to compare the part of the spectrum where we detect the targets and the part where only noise is visible. Since atmospheric targets will rarely have positive Doppler velocity with this radar configuration

(a maximum of +1 ms⁻¹ could be expected for small particles due to vertical updraft), a histogram of the full Doppler spectrum will reveal a larger number of data points at the amplitude values where the noise floor is found. This can be seen in Fig. 6b, where the noise floor, with a certain spectral width, can be easily discerned from the target echoes. By finding the amplitude corresponding to the upper edge of the noise spectral width, we can identify the maximum noise floor value and subtract it from the velocity-range spectrum. The gradual increment in the noise background at short range seen in Fig. 6b is a consequence of the close-range targets' induced phase noise and transmit-to-receive leakage. Finally, we applied Eq. (3) to obtain the final representation of data that have been made available in the form of clean reflectivity echoes in velocity-height spectra as shown in Fig. 6c.

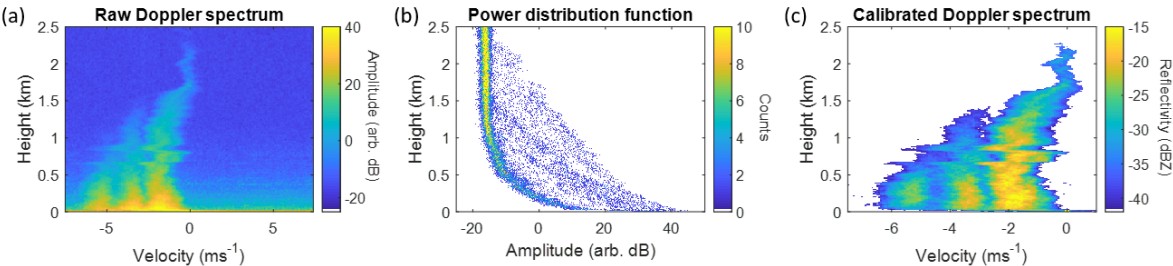

Figure 6: Processing of the G-band Doppler-range spectra (March 30 at 00:19:34 UTC): (a) Raw data as obtained from observations of target echoes showing the full 15 ms⁻¹ velocity span. (b) Noise and echo signal distribution with height. The noise floor is identified corresponding to larger number of data points at low amplitudes. (c) Final representation of the G-band Doppler spectra that are provided in the data sets described in this article.

Figure 6c represents a Doppler spectrum of an atmospheric target with different particle sizes where the echo return is spread over the range of Doppler falling velocities. While this kind of representations is particularly useful to study the particle size distribution and cloud structure at a given time, it is usually more convenient to integrate the echo returns at the different velocities and obtain the Doppler moments (Doviak and Zrnic, 1993), i.e. reflectivity, mean Doppler velocity and spectrum width, over the entire duration of the measurements.

We integrated the spectral densities that correspond to weather signals and obtained the integrated observed reflectivity as

$$Z_G = \int Z_{G,s}(v)\mathrm{d}v. \tag{4}$$

Similarly, the mean Doppler velocity and the Doppler spectrum width were calculated, respectively, as

$$\overline{v_D} = \frac{\int v\, Z_{G,s}(v)\mathrm{d}v}{\int Z_{G,s}(v)\mathrm{d}v}, \tag{5}$$

$$\sigma_{v_D} = \sqrt{\frac{\int (v - \overline{v_D})^2\, Z_{G,s}(v)\mathrm{d}v}{\int Z_{G,s}(v)\mathrm{d}v}}. \tag{6}$$

An example of the plots that can be obtained with the data sets derived from Eq. (4), (5) and (6) is shown in Fig. 7. Short-range horizontal streaks may be visible in some occurrences due to spurious artifacts originating from transmitter noise coupled into the receiver. Sporadic vertical streaks may appear as a consequence of sudden phase noise jumps, which we speculate may come from insects or birds crossing close to the radar aperture.

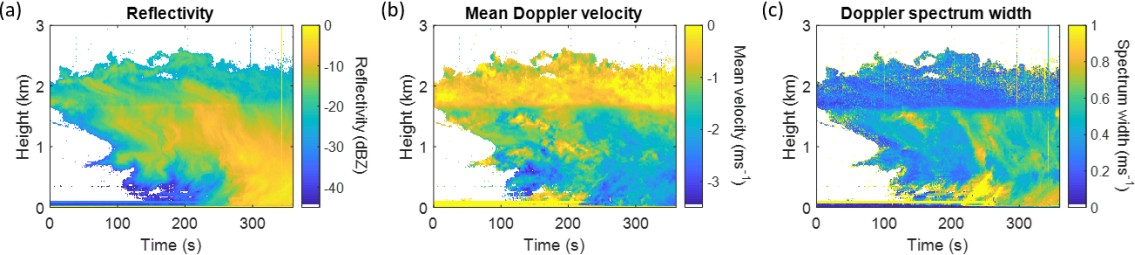

**Figure 7: (a) Reflectivity, (b) mean Doppler velocity and (c) Doppler spectrum width profiles for a cloud observation leading to surface drizzle on March 23, starting time 00:14:00 UTC. The melting layer can be easily discerned on the mean Doppler velocity and spectrum width plots at approximately 1.8 km. Visible horizontal streaks at near-zero range and close to 500 m are caused by transmitter leakage into the receiver.**

During the duration of the field campaign, we observed several occasions with simultaneous detection of low- and high-altitude targets while operating with an unambiguous range of 6.3 km. An example is shown in Fig. 8a, where low-altitude clouds are detected around 500 m. As explained in Sect. 2.3, the high-level targets, with altitudes above the 6.3 km G-band radar unambiguous range, appear in the Doppler-range spectrum folded within the first 6.3 km and are erroneously shown as low-level or mid-level signatures. We utilized the Ka-band and W-band, with much larger unambiguous range, to identify the correct altitude of the high-level targets. We then unfolded the high-level signals and corrected for the right range instead of the apparent folded range to obtain the true Doppler spectra and Doppler moments, as seen in Fig. 8b, in the cases where the low and high-level echo returns did not overlap and were distinguishable. For the occurrences where we detected clouds at precisely 6.3 km, a strong horizontal streak due to the zero-range unfolded transmit leakage will be visible.

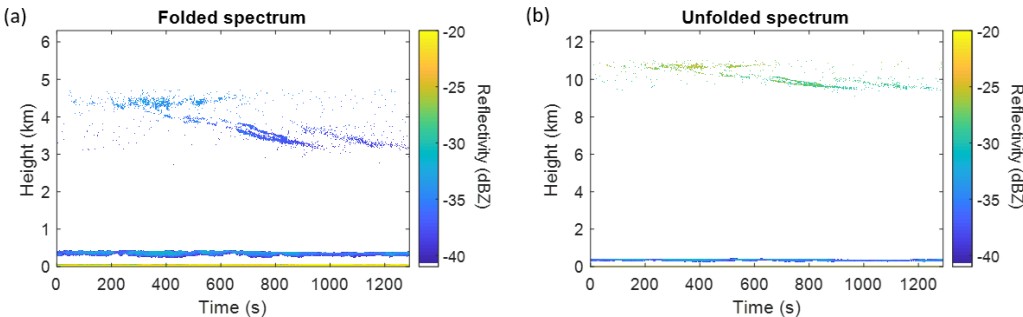

**Figure 8: (a) Folded spectrum erroneously showing high-level targets (between 9 and 11 km) as mid-level echoes (between 3 and 5 km) and (b) unfolded spectrum showing the correct altitude and reflectivity values. The plots also show low-level clouds around 500 m. The data in this figure was taken on April 11, starting time 21:22:28 UTC.**

The final G-band radar products consist of two separate data collections: one containing calibrated Doppler spectra (see Table 2 in Sect. 4), as in the example shown in Fig. 6c, and a second set with calibrated Doppler moments (see Table 3 in Sect. 4), as the ones presented in Fig. 7.

### 3.4. Ka and W-band calibration

As described in Sec. 3.2, we used a metal sphere target to calibrate the G-band radar prior to the participation in the field campaign. We followed a different approach to calibrate the Ka-band and W-band channels, using simultaneous observations of convenient cloud formations where the size of hydrometeors is much smaller than the wavelength of the transmitted signals, in such a way that the radiation is scattered following Rayleigh dispersion and effects of particle size are negligible (Lhermitte, 1990; Matrosov, 1998; Mroz et al., 2021). For reference, the transmitted wavelength of the Ka, W and G-band radars is 8.5 mm, 3.2 mm, and 1.26 mm, respectively, and hydrometeors to be used for comparison in calibration must have a diameter much smaller than those values.

We simulated the scattering behavior of liquid hydrometeors at a temperature of 280 K and number concentration of 1 $m^{-3}$ for different drop sizes and the three transmitted frequencies using Python's open source PyMieScatt package (Sumlin et al., 2017). As seen in Fig. 9, the effects of particle size on radiation dispersion begin to be noticeable for drop diameters larger than 0.3 mm at 238.8 GHz and 0.7 mm at 94.88 GHz.

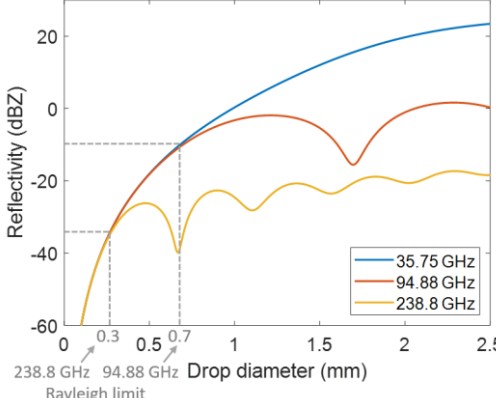

**Figure 9: Reflectivity as a function of the drop diameter for liquid spheres at 280 K and a number concentration of 1 $m^{-3}$. The drop diameter limits for Rayleigh scattering at 238.8 and 94.88 GHz are highlighted.**

A radar cannot directly measure the drop diameter during observations, but CloudCube's G-band instrument is able to retrieve the Doppler velocity of the hydrometeors. We can use this capability to relate the measured Doppler fall velocity with the drop diameter, as has extensively been studied in the literature (Du Toit, 1967; Atlas et al., 1969), and estimate a drop fall velocity limit at which we should calibrate the Ka and W-band radars.

The equilibrium between the downwards gravitational force and the upward aerodynamic drag determines the terminal fall velocity of hydrometeors. This velocity depends, among other parameters, on the cross-sectional area of the hydrometeors, their volume, and the medium density. A common approximation to derive the drop fall terminal velocity is to use an empirical formulation that expresses the velocity in terms of the drop diameter as (Atlas et al., 1969)

$$v = 9.65 - 10.43e^{-0.6d}, \tag{7}$$

where $d$ is the drop diameter in millimeters.

Figure 10 is used to illustrate the relationship in Eq. (7), where we can see how the hydrometeor diameter limits for the Rayleigh scattering regime, indicated in Fig. 9, correspond to drop fall terminal velocities of approximately -1 ms$^{-1}$ (0.3 mm diameter) and -3 ms$^{-1}$ (0.7 mm diameter).

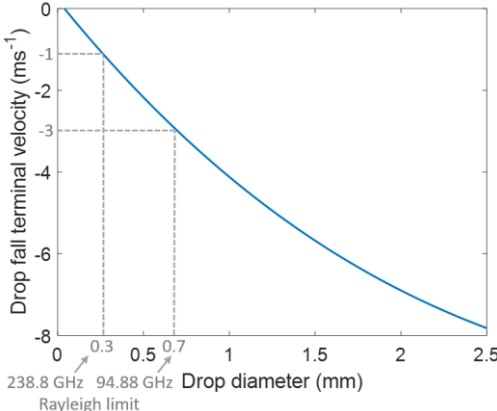

**Figure 10: Relationship between the drop fall terminal velocity and the drop diameter following the formulation in Atlas et al. (1969). The drop fall velocity limits for Rayleigh scattering at 238.8 and 94.88 GHz are highlighted.**

As a first approximation, we can assume that the hydrometeors are in vertical dynamic equilibrium, and that the population of particles contained within the G-band radar volume resolution are all the same size (small Doppler spectrum width). 
We can then use the measured mean Doppler velocity equivalently to the drop fall velocity, and estimate the diameters of the falling hydrometeors as a function of range. Therefore, we can evaluate the regions where the cross-calibration can be performed by taking advantage of the information provided by the G-band Doppler velocity plots.

Over the duration of the field campaign, we observed formations with suitable Doppler velocities on different days to identify the echo signals where we could perform the intercalibration and also to confirm the consistency and validity of the 
method among different cases. An example of a low-level stratocumulus that we used to cross-calibrate the instruments is shown in Fig. 11a. We converted the mean Doppler velocity data into particle diameter information using Eq. (7) (see Fig. 11b), in order to discern the regions where the signals had been scattered following Rayleigh dispersion (shown in Fig. 11c).

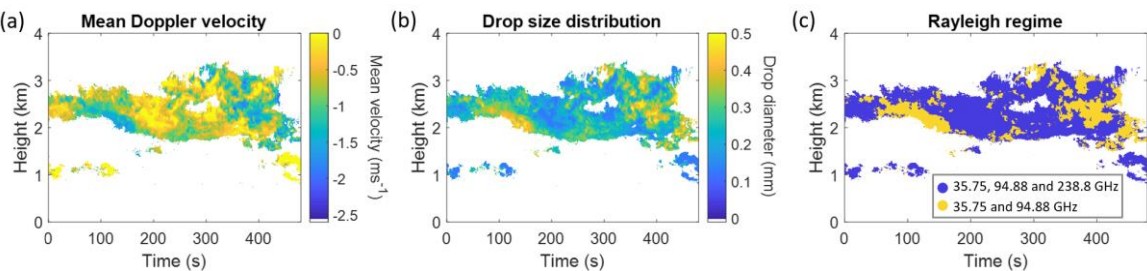

**Figure 11: Example of a cloud formation on March 30, starting time 21:27:26 UTC, selected to perform the intercalibration. (a) The mean Doppler velocity of the hydrometeors is obtained from the G-band radar measurements as explained in Sect. 3.3. (b) The drop diameter profile is derived from (a) after applying Eq. (7). (c) Rayleigh scattering regions are differentiated based on (b) according to the radars transmit frequency. The blue areas correspond to particle diameters below the Rayleigh limit at 35.75, 94.88 and 238.8 GHz whereas the yellow parts discern particle sizes where only the 35.75 and 94.88 GHz dispersion is below the Rayleigh limit.**

Once the optimal observations and regions to cross-calibrate the Ka and W-band were identified, we deduced a calibration factor for the Ka-band and W-band channels that we applied to the rest of observations. Based on the previously calculated G-band calibration factor, we determined the W-band correction as

$$C_W = C_G \frac{\lambda_W^4 e^{-2\beta_W} |K_G|^2 \Omega_G \Delta r_G}{\lambda_G^4 e^{-2\beta_G} |K_W|^2 \Omega_W \Delta r_W} G_{PC,W} \frac{P_G}{P_W}, \qquad (8)$$

where $P_G/P_W$ is the ratio of the G-band to W-band echo amplitudes at the locations where the signals are scattered following Rayleigh dispersion (blue areas shown in Fig. 11c). The term $G_{PC,Ka/W}$ accounts for the pulse compression gain of the Ka and W-band systems that can be obtained from the pulse width $\tau$ and the chirp bandwidth $B$ as $G_{PC} = \tau B$.

We found good agreement between the different cases used to intercalibrate the W-band instrument based on the G-band radar calibration. However, the Ka and G-band cross-calibration showed higher discrepancy among the diverse scenarios, likely due to the huge gap in transmitted wavelengths and the different radar sensitivities (see Table 1) that made it difficult to obtain echo returns from exactly the same hydrometeors. We learned, however, that by comparing the echo returns from the Ka and W-band radars outside of the G-band Rayleigh region, i.e. the brighter returns corresponding to larger particles in the yellow areas in Fig. 11c, the agreement was substantially improved. We, therefore, used the W-band radar to intercalibrate the Ka-band instrument and obtain the Ka-band calibration factor as

$$C_{Ka} = C_W \frac{\lambda_{Ka}^4 e^{-2\beta_{Ka}} |K_W|^2 \Omega_W \Delta r_W}{\lambda_W^4 e^{-2\beta_W} |K_{Ka}|^2 \Omega_{Ka} \Delta r_{Ka}} \frac{G_{PC,Ka}}{G_{PC,W}} \frac{P_W}{P_{Ka}}. \qquad (9)$$

Since the W-band and Ka-band modules calibration is based on the G-band radar absolute calibration, the uncertainty in determining the W-band and Ka-band calibration factors primarily inherits the 1 dB error discussed in Sect. 3.2.

## 3.5. Ka and W-band reflectivity profiles

The Ka and W-band systems provide the echo power of any given target as a function of range. Once the calibration factors are calculated following the analysis described in Sect. 3.4, observed reflectivity profiles can be obtained from echo power measurements as

$$Z_{Ka/W} = C_{Ka/W} r^2 P_{Ka/W},$$
(10)

with $r$ being the target range.

Figures 12a and 12d show an example of Ka and W-band data, respectively, as obtained during measurements, after averaging 256 collected pulses. In a similar approach as for the G-band data (see Sect. 3.3), we studied the distribution of the echo returns to identify and subtract the noise level of the Ka and W-band instruments and produce cleaner and higher-quality data sets. By making histograms including the range where the target signals are not present, as plotted in Fig. 12b and 12e, we determined the amplitude value to be subtracted that corresponds to the upper edge of the noise background spectral width. For the case of the W-band observations (see Fig. 12d), we can see a close-range area with high amplitude values. This comes from the zero-range calibration pulse and we have removed this region from the data sets. We can also observe a close-range bright region in the Ka-band spectrum in Fig. 12a. This signal extends to altitudes slightly higher than the blind range of the W-band radar although it is not noticeable there. We also did not see such echoes in the range-Doppler spectrum of the G-band instrument due to its implementation as a FMCW radar. These artifacts are likely due to transmit-to-receive leakage as a consequence of the bistatic configuration of the Ka-band radar. We have also discarded the data points corresponding to these artifacts to compile the final data sets.

Once the noise floor and artificial echoes have been subtracted, we used Eq. (10) to calculate the observed calibrated reflectivity profiles as shown in Fig. 12c and 12f.

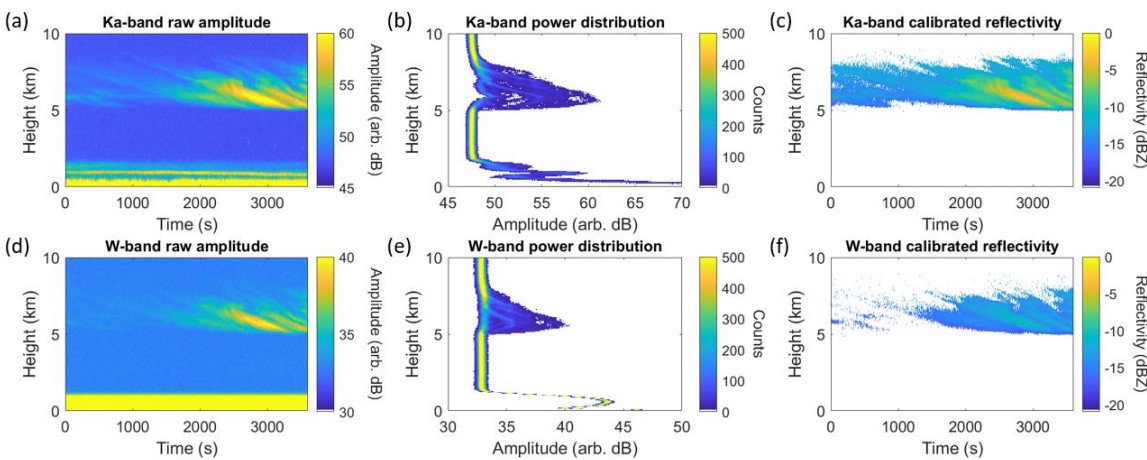

**Figure 12: Simultaneous measurements at Ka and W-band for a mid/high-level formation on March 31, starting time 19:43:24 UTC. (a) and (d) The echo detections are received in the form of range-amplitude spectra. (b) and (e) Data power distributions are used to determine the noise floor and identify artificial echoes. (c) and (f) After cleaning the spectra, Eq. (10) is used to calculate the reflectivity profiles.**

## 3.6. Multifrequency reflectivity and dual-ratio reflectivity profiles

While reflectivity profiles can provide information about the hydrometeor content in a particular atmospheric formation, the combination and simultaneous analysis of multiple frequencies can reveal valuable insights into particle size distributions. This is well understood after deriving reflectivity ratios between the different frequencies, where the resulting ratio profiles reveal the scattering properties and differential attenuation of the hydrometeors, which is directly related to the size and spatial distribution of such particles inside the cloud envelope.

To be able to focus the study on the liquid and ice hydrometeors, it is important to subtract the contribution of gaseous attenuation in the atmosphere, which also has a frequency-dependent behavior. We used for such purpose the data obtained from radiosondes that were released in a daily basis, every six or twelve hours, next to the location where CloudCube operated. The radiosondes measured the temperature, pressure, and relative humidity with height, that we utilized to calculate the two-way gaseous attenuation correction using the model of Rosenkranz et al., 1998, to apply to our radar measurements.

The three CloudCube modules were operated independently during the deployment, and the recording periods were manually set. The first step to jointly process the data was to synchronize the time stamps for every frequency channel. After converting the data time stamps in every radar to Universal Coordinated Time (UTC), we selected the latest starting time and the earliest end time among the three radars data sets for comparison to set the temporal limits to process the data in conjunction. Then, we linearly interpolated the collected data (previously converting the echo returns to linear units) to match the least common multiple between the different temporal resolutions of the three instruments. Besides finding a common temporal axis, we also needed to match the spatial resolution of the instruments. The Ka-band and W-band radars were operated with a sampling resolution of 60 m, while the range resolution of the G-band instrument was 10 m as described in Table 1. To also obtain a common spatial resolution, we integrated the G-band echo returns from the 10 m resolution cells over 60 m. Once the data from the three instruments are spatiotemporally matched, we can now compare and study the relationship between the hydrometeors frequency-dependent echo returns. We applied the following expressions to calculate the dual-frequency reflectivity ratios between the three possible combinations

$$DFR_{Ka-W} = Z_{Ka}/Z_W, \tag{11}$$

$$DFR_{Ka-G} = Z_{Ka}/Z_G, \tag{12}$$

$$DFR_{W-G} = Z_W/Z_G. \tag{13}$$

The resulting reflectivity and dual-frequency ratio plots with matching temporal and spatial resolutions, and gaseous attenuation subtracted, are shown in Fig. 13 for an example case. These data products have also been made available (see Table 4 in Sect. 4).

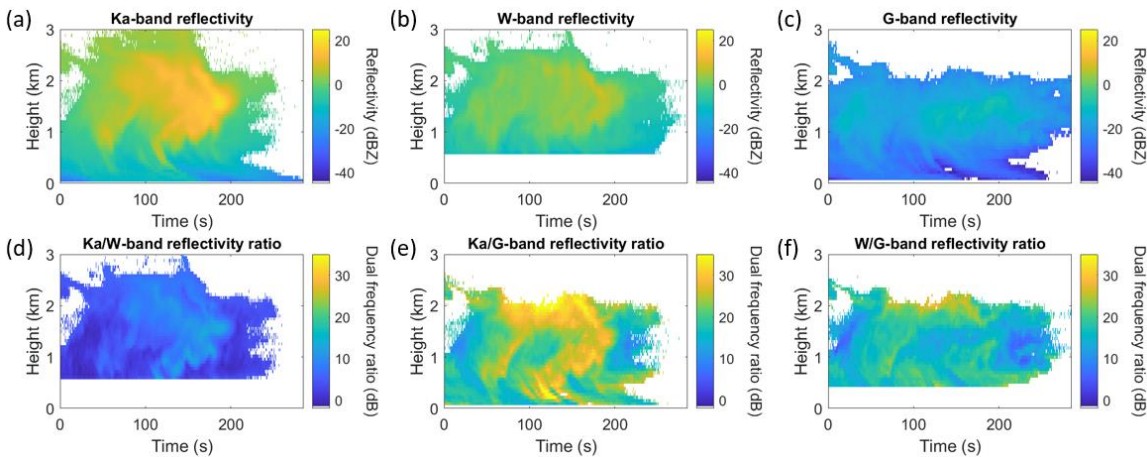

**Figure 13: Reflectivity and dual-frequency ratio plots for the Ka, W and G-band frequencies for a stratocumulus formation on March 30, starting time 18:15:44 UTC. Note that the blind close-range from the W-band instrument has been subtracted, as explained in Sect. 3.5, which has an impact on the Ka/W-band and W/G-band differential attenuation plots, limiting the information at close-range.**

**4. Data availability**

The data for the three CloudCube modules described in this article are provided in netCDF format in the following packages at https://doi.org/10.5281/zenodo.10076227 (Socuellamos et al., 2024):

- The G-band Doppler spectra, as the one shown in Fig. 6c, can be found at in a package under the name *CloudCube_EPCAPE_Gband_Spectra.zip*. The data inside the folder are sorted separately for each day and time of
operation in the format YYYMMDD_HHMMSS (year:month:day_hour:minute:second) where HHMMSS corresponds to the starting time of operation in UTC of the particular data set. The content of the .nc files consist of six variables (listed in Table 2): the starting time of the observation referenced to Unix epoch (*base_time*) in units of seconds (s) since 1970-01-01 00:00:00 UTC, the temporal extent of the measurement volume in seconds (s) since volume start (*time_offset*) and since epoch (*time*), the distance to the targets (*range*) in meters (m), their Doppler
velocity (*velocity*) in units of meters per second (ms$^{-1}$), and the targets' spectral reflectivity (*reflectivity*) in units of decibels relative to reflectivity (dBZ).

- The G-band data containing the Doppler moments, i.e. reflectivity, Doppler mean velocity, and Doppler spectrum width, with an example shown in Fig. 7, have been uploaded in a package named *CloudCube_EPCAPE_Gband_Moments.zip* (see Table 3). The data is sorted for each day and time of operation in
the format YYYYMMDD_HHMMSS. The netCDF files contain the variables *base_time*, *time_offset*, *time* and *range* as described in the previous bullet point, the reflectivity (*reflectivity*) in units of decibels relative to reflectivity (dBz),

and the mean Doppler velocity (*mean_doppler_velocity*) and Doppler spectrum width (*spectral_width*) in units of meters per second (ms$^{-1}$).

- The Ka, W and G-band reflectivities and dual-frequency reflectivity ratios, with matching temporal and range resolutions and gaseous attenuation subtracted, are provided in the folder *CloudCube_EPCAPE_Multifrequency.zip*. Ten variables (described in Table 4) can be found in the netCDF files that are sorted by the different days and times of operation: *base_time*, *time_offset*, *time* and *range* as described in the first bullet point, reflectivity in decibels relative to reflectivity (dBZ) for each frequency band (*reflectivity_ka, reflectivity_w,* and *reflectivity_g*) and dual frequency reflectivity ratio in decibels (dB) for the three possible combinations between the frequencies of operation (*dual_frequency_ratio_ka_w, dual_frequency_ratio_ka_g,* and *dual_frequency_ratio_w_g*).

Table 2, 3 and 4 summarize the three different data sets, the variables and the files that have been made available. The size of the variables is given by the combination of the *range*, *time*, and *velocity* dimensions. Missing values in the data variables are filled with NaN.

**Table 2: Description of the files and variables included in the G-band Doppler spectra data package available to download.**

| Link | https://doi.org/10.5281/zenodo.10076227 |
|------|------|
| Package folder | CloudCube_EPCAPE_Gband_Spectra.zip |
| Files | YYYYMMDD_HHMMSS_Gband_Spectra.nc |

| Variable name | Dimensions | Units | Long name |
|------|------|------|------|
| *base_time* | - | s | Base time in Epoch |
| *time_offset* | time | s | Time in seconds since volume start |
| *time* | time | s | Time in seconds since Epoch |
| *range* | range | m | Radial range to measurement volume |
| *velocity* | velocity | ms$^{-1}$ | Radial Doppler velocity |
| *reflectivity* | range, velocity, time | dBZ | Spectral equivalent reflectivity factor |

**Table 3: Description of the files and variables included in the G-band moments data package available to download.**

| Link | https://doi.org/10.5281/zenodo.10076227 |
|------|------|
| Package folder | CloudCube_EPCAPE_Gband_Moments.zip |
| Files | YYYYMMDD_HHMMSS_Gband_Moments.nc |

| Variable name | Dimensions | Units | Long name |
|------|------|------|------|
| *base_time* | - | s | Base time in Epoch |

| Variable name | Dimensions | Units | Long name |
|---|---|---|---|
| *time_offset* | time | s | Time in seconds since volume start |
| *time* | time | s | Time in seconds since Epoch |
| *range* | range | m | Radial range to measurement volume |
| *reflectivity* | range, time | dBZ | Equivalent reflectivity factor |
| *mean_doppler_velocity* | range, time | ms$^{-1}$ | Radial mean Doppler velocity |
| *spectral_width* | range, time | ms$^{-1}$ | Spectral width |

**Table 4: Description of the files and variables included in the multifrequency data package available to download.**

| | |
|---|---|
| **Link** | https://doi.org/10.5281/zenodo.10076227 |
| **Package folder** | CloudCube_EPCAPE_Multifrequency.zip |
| **Files** | YYYMMDD_HHMMSS_Multifrequency.nc |

| Variable name | Dimensions | Units | Long name |
|---|---|---|---|
| *base_time* | - | s | Base time in Epoch |
| *time_offset* | time | s | Time in seconds since volume start |
| *time* | time | s | Time in seconds since Epoch |
| *range* | range | m | Radial range to measurement volume |
| *reflectivity_ka* | range, time | dBZ | Equivalent reflectivity factor Ka-band |
| *reflectivity_w* | range, time | dBZ | Equivalent reflectivity factor W-band |
| *reflectivity_g* | range, time | dBZ | Equivalent reflectivity factor G-band |
| *dual_frequency_ratio_ka_w* | range, time | dB | Dual frequency ratio Ka/W-band |
| *dual_frequency_ratio_ka_g* | range, time | dB | Dual frequency ratio Ka/G-band |
| *dual_frequency_ratio_w_g* | range, time | dB | Dual frequency ratio W/G-band |

**5. Code availability**

The processing codes can be made available upon request to the corresponding author.

**6. Conclusion**

CloudCube, a new multifrequency radar to profile atmospheric phenomena, participated in the EPCAPE field campaign during six weeks in the months of March and April 2023, with a focus on measuring marine structures to study their formation and evolution. A variety of cloud formations were observed during that period, obtaining a wide and ample data

collection comprising observations on different days that can be used to analyze the microphysics and dynamics of such processes.

This article introduced the different data sets that have been made available after implementation of a selection and data-quality control process. Doppler moments and spectra data have been provided for the G-band module, while

multifrequency reflectivity and dual-frequency ratios data are accessible at Ka, W and G-band. These data sets contain the first atmospheric observations at 238.8 GHz, making this an exceptional collection never offered before.

Simultaneous observations at different frequency bands including the G-band, such as the ones CloudCube perform, can reveal the size and distribution of drops with diameters in the millimeter and submillimeter range from the differential scattering and attenuation properties of the hydrometeors, and fill important observational gaps to improve cloud-climate

feedback and aerosol-cloud interaction models.

**Author contributions**

MDL coordinated the participation in EPCAPE. RRM led the development of CloudCube. KBC and RMB developed CloudCube's digital processors. All the authors collectively contributed to preparing, installing and operating the radars for and during the field campaign. JMS and AU processed and prepared the datasets. JMS composed the manuscript with

contributions from the rest of the authors.

**Competing interests**

The authors declare that they have no conflict of interest.

**Acknowledgements**

This research was supported by the National Aeronautics and Space Administration Earth Science Technology Office

(NASA ESTO) under the Instrument Incubator Program, and carried out at the Jet Propulsion Laboratory, California Institute of Technology, under a contract with NASA (80NM0018D0004).

Radiosonde data were obtained from the atmospheric radiation measurement (ARM) user facility, a U.S. department of energy (DOE) office of science user facility managed by the biological and environmental research program.

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
