# Peer review of "Multifrequency radar observations of marine clouds during the EPCAPE campaign"

_Earth System Science Data, 2023_

## Author Comment (AC1)

Dear Referee,

We would like to thank you for the very useful suggestions to improve the quality of the manuscript. Please see below our response to the comments.

In addition, we have made additional minor modifications to improve the overall readability. These include: adopting the standard terminology of Doppler spectra and Doppler moments to refer to the final G-band radar data products, using the universal date format (YYYYMMDD_HHMMSS) to name the data files, and adding new references to provide more context to the analysis performed in this work.

All the changes are highlighted in the revised manuscript.

Sincerely,
The Authors

**General Comments**:

This manuscript presents a new dataset containing triple-frequency, vertical-pointing radar (CloudCube) data during the EPCAPE field campaign along the California coastline. This dataset is both unique and useful in that it contains co-located G-band observations (238.8 GHz) with lower frequency channels (Ka- and W-band). Importantly, these are the first atmospheric observations at this frequency (238.8 GHz), which can detect much smaller hydrometeors (cloud and precipitation) than the Ka- and W-band radars. Though I'm unfamiliar with calibration techniques for radars, the explanation is clear and sufficient. The dataset itself is easily accessible on Zenodo and complete, with any missing data explained (Fig 3). This dataset can help inform on choosing the appropriate radar frequencies for future field campaigns or satellite missions, such as the Atmosphere Observing System (AOS) mission. I recommend the manuscript be accepted with minor revisions for clarity.

**Specific Comments:**

L9-10: Though it's implicit, it would be helpful to specifically call the radar "ground-based" at some point early on. The name CloudCube initially gave the impression that this could be a spaceborne instrument.

We have added "*ground-based*" when we introduce CloudCube in the abstract, the introduction, and Table 1 (lines 9, 58 and 93).

Fig. 5 caption and L211-213: "short-range horizontal streaks" and "sporadic vertical streaks": It's difficult to see what features are being pointed out. Perhaps describe where in these plots (time, height)

We have modified Fig. 7 (previously Fig. 5) caption to indicate the position of these streaks (lines 254-255):

*"Visible horizontal streaks at near-zero range and close to 500 m are caused by transmitter leakage into the receiver."*

L71,140 and elsewhere: the use of "range" to describe the height above the radar is somewhat confusing. I think explicitly stating somewhere that range = distance from radar would be helpful to better understand "blind range," "close-range," "unambiguous range," etc.

We have added the following sentence in Sect. 2.2. lines 101-102:

*"Since the radars were pointing zenith in this configuration, we have used range and height interchangeably in this manuscript to describe the targets' distance to the radars."*

**Technical Corrections:**

L38: change "this kind" to "these kinds": "...suitable fit for these kinds of measurements…"

The recommendation has been accepted (line 38).

L287: "...blue areas respond to particle diameters below the Rayleigh limit…" should this be "blue areas correspond"?

The recommendation has been accepted (line 322).

---

## Author Comment (AC2)

Dear Referee,

We would like to thank you for the very useful suggestions to improve the quality of the data sets and the manuscript. Please see below our response to the comments.

In addition, we have made additional minor modifications to improve the overall readability. These include: adopting the standard terminology of Doppler spectra and Doppler moments to refer to the final G-band radar data products, using the universal date format (YYYYMMDD_HHMMSS) to name the data files, and adding new references to provide more context to the analysis performed in this work.

All the changes are highlighted in the revised manuscript.

Sincerely,
The Authors

**General Comments:**

The paper by Socuellamos et al., describes a new multifrequency radar dataset that has important implications for future remote sensing precipitation missions. The dataset appears to be of high quality, and is unique in that it offers the first look into clouds using a high frequency G-band radar. These types of instruments are incredibly powerful for viewing very light intensity precipitation that is otherwise unobservable using traditional radar systems. The fact that these observations are combined with other, more common frequency radars, opens the door to exciting new possibilities in precipitation research. Overall, the paper is well written, and doesn't contain any major technical flaws. While this is an important dataset to release to the public, and the topic is certainly relevant to the readers of ESSD, there are some improvements to the dataset itself (primarily related to documentation, missing data choices and data artifacts) that I'd like to see improved upon before I can fully recommend this for publication. Further, due to the different radars used here, and the multiple quality assurance (QA) and calibration techniques that were applied, this manuscript would greatly benefit from a data processing diagram to summarize the entire process in a digestible manner. With these changes, I believe the paper would be an excellent addition to ESSD.

**Major Comments:**

1. The most substantial comment I have for this paper is related to the general structure, formatting and description of the dataset itself (which is the focus/primary product from this work). While the data contents are certainly useful, its current state could be improved and made more accessible by following CF Metadata Conventions, and by addressing a few of the remaining data artifacts I encountered. For a detailed list of these comments, please see my specific notes in "*Dataset Comments*" below.

   We have made substantial changes on the data sets and modified the data availability section (Sect. 4) accordingly. Please see below for a detailed response to the specific comments.

2. Further, there was a fair bit of QA done to improve the quality of the final datasets. This is great, but when combined with the fact that there are multiple different radars here, it can be challenging as a reader to follow what exactly was done to which product. It would be beneficial to provide another figure early on that summarizes the data processing and QA steps. This is something that you could refer back to throughout the document to make the process clearer. I add this as a major comment, as it might be a bit of work to concisely condense all of this information within a single figure, but I believe it would be a valuable contribution.

We have added a new section (Sect. 3.1) and Fig. 5 (Lines 164-180):

"*3.1. Overview*

*The final data products that are described in this article have gone through several steps to provide calibrated reflectivity and to enhance the overall quality of the data sets. A flowchart of the process illustrating the different steps followed to obtain the final data products is shown in Fig. 5. Initially, we applied a data quality control process that included selecting relevant observations, removing noise and artifacts, and, in the case of the G-band data, unfolding the G-band Doppler spectra where possible (step 1). We then applied a calibration factor to the G-band Doppler spectra data (2), previously obtained from an absolute calibration of the radar, to obtain calibrated spectral reflectivity and form the first data product (3). Subsequently, we calculated the G-band Doppler moments (4), which constitute the second data product discussed in this article. Finally, we utilized the G-band Doppler moments to identify optimal atmospheric formations to cross-calibrate the W-band and Ka-band raw data using the G-band absolute calibration as reference (5). After spatiotemporally matching the calibrated data and subtracting the gaseous attenuation at the three different frequency bands, we produced the third and final data product which includes multifrequency reflectivity and dual-frequency reflectivity ratios (6). The different steps in CloudCube's data processing are described in more detail in the following subsections.*"

[Figure]

*Figure 5: CloudCube's data processing flowchart. A data quality control and calibration process were applied to the raw data to produce three separate data sets: G-band Doppler spectra, G-band Doppler moments, and Ka, W, and G-band multifrequency reflectivity and dual-frequency ratios.*

**Dataset Comments:**

Each of these comments are in reference to the provided NetCDF datasets available for download on Zenodo.

1. **Standardization:** I would recommend rewriting the data variables in the NetCDF files using CF Metadata Conventions where possible (https://cfconventions.org/). Using standard properties like *standard_name*, *long_name*, *missing_val* within the file metadata makes accessing and dealing with the data much easier for consumers.

   We have adopted CF Metadata Conventions to rewrite the data variables. Please check the data files. Here is an example of the *reflectivity_ka* variable in 20230330_162540_Multifrequency.nc:

   reflectivity_ka
          Size:     100x611
          Dimensions: range, time
          Datatype:   single
          Attributes:
                _FillValue   = NaN
                long_name   = 'Equivalent reflectivity factor - Ka band'
                units      = 'dBZ'
                standard_name = 'equivalent_reflectivity_factor'

2. **Data Artifacts:** I noticed an odd, unphysical looking band of reflectivity in the G-band data on the following days: *040123_004927_Multifrequency.nc, 033123_194324_Multifrequency.nc, 033123_224802_Multifrequency.nc, 033123_234344_Multifrequency.nc.* Do you have some insight into what is going on in these cases? I wonder if there is a way to mask these regions, as they might lead to further derived issues later on (e.g., you can see problems in the reflectivity ratios too). Perhaps these issues are discussed in the paper and I just missed it?

   This horizontal band is a consequence of the zero-range transmitter leakage that appears at 6.3 km when we unfold the Doppler spectra. We had discussed this issue in the original version of the manuscript at the end of Sect. 3.2. (now Sect. 3.3, lines 264-265):

   "… *For the occurrences where we detected clouds at precisely 6.3 km, a strong horizontal streak due to the zero-range unfolded transmit leakage will be visible.*"

3. **Missing Values:** I also noticed an issue with the missing values prescribed on the following days: *041323_232910_Multifrequency.nc, 041323_235527_Multifrequency.nc, 041423_000835_Multifrequency.nc, 041423_002144_Multifrequency.nc, 041423_003452_Multifrequency.nc, 041423_004800_Multifrequency.nc.* It appears that when the blind zone was being masked, instead of NaNs, -inf values were assigned. I would recommend sticking to a single *missing_val* type (NaN), as -inf values are quick to saturate data scales and make plots generally unreadable.

We have used NaN consistently across all data sets.

4. **Timesteps:** I would also recommend avoiding non-standard time steps. It can be a bit of a hassle to implement this, but it saves users a lot of time trying to figure out when time step 0 starts in a file (reading that info from the filename is lost if ever stripped/renamed). Ideally, the time axis would have a UTC variable that begins at the same point for each file (e.g. midnight), and would have exactly 86400 time steps (seconds in a day) with missing NaN columns.

All data files have now three "time" variables (*base_time*, *time_offset*, *time*) to provide an absolute and relative temporal axis for each data set. We have adopted the standard Unix epoch (1970-01-01 00:00:00 UTC) as the reference base time for the data sets.
- The variable *base_time* defines the starting time of each data set, in seconds, since Unix epoch. Therefore, the starting time of each file is now included as a data variable, and no longer relies on preserving the filename.
- The variable *time* provides an absolute temporal range, in seconds, referenced to Unix epoch. Hence, every data set now contains a common temporal reference and axis.
- Finally, the variable *time_offset* provides a relative temporal range, in seconds, referenced to the data set starting time.

A similar comment for vertical extent, why do some files (e.g., 041123_220747_Multifrequency.nc) have a larger vertical extent than others? Shouldn't this be set to a fixed value for consistency? If it is to save on file size, you could also deflate the NetCDFs considerably (e.g., deflate level 2) as the float64 double you are using here is likely excessive in terms of required precision for these observations.

We have adopted 6 km as the fixed vertical extent for the majority of the files. For the few cases where we needed to unfold the G-band spectra, we have increased the vertical extent to 12 km.

We have deflated and changed the precision of some of the variables to considerably reduce the file sizes.

5. **Dimension Variables:** I don't think the height and time variables are correctly set as data dimensions (files are currently using Nr and Nt). To be clear, it is fine to have those variables as dimensions, but they are unitless in this case, and data users shouldn't have to go look at another variable to check what the heights are (i.e., you can package that together into a single dimensional variable in the NetCDF).

The data dimensions are now *time*, *range* and, for the G-band Doppler spectra, also *velocity*.

6. **Metadata:** This is sort of related to point 1 on CF-conventions, but the variable metadata is generally lacking in detail, and there is no contact information in the global attributes.

Ideally you want your dataset to be able to somewhat stand on its own without the associated manuscript and I would add these descriptors for that reason.

We have added several global attributes to improve the description of the data sets. Please check the data files. Here is an example of the global attributes in 20230330_162540_Multifrequency.nc:

```
location_description        = 'Eastern Pacific Cloud Aerosol Precipitation Experiment
(EPCAPE), Scripps Pier, La Jolla, CA'
institution        = 'NASA Jet Propulsion Laboratory (JPL), California Institute of
Technology (Caltech)'
instrument_name        = 'CloudCube'
title                = 'JPL CloudCube Multi-frequency Reflectivities and Ratios'
doi                = '10.5281/zenodo.10076227'
history        = 'created by arturo.umeyama@jpl.nasa.gov on 2024-04-04 18:14:51 UTC'
frequency_ka            = '35.75 GHz'
frequency_w            = '94.88 GHz'
frequency_g            = '238.8 GHz'
transmission_type_ka        = 'Pulsed'
transmission_type_w        = 'Pulsed'
transmission_type_g        = 'FMCW'
pulse_width_ka            = '1 us'
pulse_width_w            = '1 us'
pulse_width_g            = '40 us'
pulse_repetition_interval_ka = '2.000 ms'
pulse_repetition_interval_w  = '1.000 ms'
pulse_repetition_interval_g  = '0.042 ms'
chirp_bandwidth_ka        = '0 MHz'
chirp_bandwidth_w        = '0 MHz'
chirp_bandwidth_g        = '15 MHz'
peak_transmit_power_ka    = '10 W'
peak_transmit_power_w     = '10 W'
peak_transmit_power_g     = '0.24 W'
antenna_diameter_ka    = '30 cm'
antenna_diameter_w     = '30 cm'
antenna_diameter_g     = '60 cm'
unambiguous_range_ka    = '300.00 km'
unambiguous_range_w     = '150.00 km'
unambiguous_range_g     = '6.30 km'
range_resolution_ka    = '150 m'
range_resolution_w     = '150 m'
range_resolution_g     = '10 m'
range_sampling        = '60 m'
```

**Minor Comments:**

1. Can the authors comment on the applicability of the G-band radar for very light intensity snowfall? This technology seems incredibly powerful for observations of fine ice crystals, for instance.

   We have added the following paragraph in the introduction (Sect. 1, Lines 43-50):

   "*…including the G-band, can be used to characterize particle size distributions with drop sizes in the submillimeter range, and to detect small amounts of liquid water content, revealing new valuable information about cloud and precipitation behavior (Battaglia et al., 2014). In addition, the combination of G-band Doppler radar with lower frequency channels offers significant benefits for quantifying the properties of ice-phase hydrometeors. As suggested by Battaglia et al. (2014), using dual-frequency reflectivity ratios from three different channels including G-band has the potential to identify snow crystals habit, while Hogan et al. (2000) point out the utility of the G-band dual-frequency ratio for sizing cirrus crystals. With the burgeoning availability of multifrequency radar observations including G-band (Lamer et al., 2021, Courtier et al., 2022), the coming years offer a tremendous opportunity to validate these theorized remote sensing capabilities.*"

   And added the references:

   Battaglia, A., Westbrook, C. D., Kneifel, S., Kollias, P., Humpage, N., Löhnert, U., Tyynelä, J., and Petty, G. W.: G band atmospheric radars: new frontiers in cloud physics, Atmos. Meas. Tech., 7, 1527–1546, https://doi.org/10.5194/amt-7-1527-2014, 2014.

   Hogan, R. J., Illingworth, A. J., and Sauvageot, H.: Measuring Crystal Size in Cirrus Using 35- and 94-GHz Radars, J. Atmos. Ocean Tech., 17, 27–37, https://doi.org/10.1175/1520-0426(2000)017<0027:MCSICU>2.0.CO;2, 2000.

2. While I realize that the experiment lasted 6 weeks, there are really only 6 days of comparable observations across all radars. I think this fact should be brought up earlier in the paper, as I was expecting there to be much more data than what really exists for all three radars.

   We have added "*on six different days*" when we first discuss the multifrequency data in the abstract and the introduction (Lines 18 and 65).

3. What are the vertical extents for each of the radar instruments? Is this provided somewhere because it wasn't clear to me from the information in Table 1?

   The vertical extent of the radars depends on the radar parameters and the target's characteristics. During this field campaign, we were able to detect targets at altitudes close to 12 km.

4. The Figure 3 colors make this challenging to read if you are colorblind. I would recommend changing the palette here for accessibility. I would also add a bit more space between hatches on the '*Data available but not provided*' for further clarity.

We have changed the colors in Fig. 3 and increased the space between hatches:

[Figure]

We have also changed the color of one of the curves in Fig. 9:

[Figure]

5. Figure 5 is really neat, what is this horizontal feature at about 1.75 km? Also, what date/time is this occurring at? I would include the date/time in the figure somewhere.

We have added the following sentence in Fig. 7 (previously Fig. 5) caption (lines 253-254):

"*The melting layer can be easily discerned on the mean Doppler velocity and spectrum width plots at approximately 1.8 km.*"

We have added the date and time in the caption of all figures showing observations.

6. The process for Ka and W-band calibration in Section 3.3 is quite interesting. Is this a fairly standard procedure?

(Now Sect. 3.4) Similar procedures have been used in the past to cross-calibrate multifrequency radar systems using Rayleigh-scattering regions. We have added a new reference to the previously already mentioned in line 280:

Matrosov, S. Y.: A Dual-Wavelength Radar Method to Measure Snowfall Rate, J. App. Meteor., 37(11), 1510-1521, https://doi.org/10.1175/1520-0450(1998)037<1510:ADWRMT>2.0.CO;2, 1998.

I noticed you mention different assumptions (i.e., "*we can assume that the hydrometeors are in vertical dynamic equilibrium, and that the population of particles contained within the G-band radar volume resolution are all the same size*"), and this process must therefore have some associated uncertainty wrt. the calibration that I feel should be discussed.

Note that these assumptions are related to identifying the proper regions to perform the intercalibration and do not necessarily affect the accuracy of the method. Since we used different days and atmospheric formations to validate the technique, we believe that additional uncertainty coming from these assumptions is negligible.

We have added the following paragraph in Sect. 3.2., lines 196-200:

"*Uncertainties in the determination of the calibration factor may arise from an inaccurate knowledge of the radar parameters and weather conditions needed as input values in Eq. (1), or from an imperfect alignment of the calibration sphere to the radar beam center. While these uncertainties are difficult to quantify precisely, Roy et al. (2020) estimate that they may lead to an error of around 1 dB in the final calibrated reflectivity values.*"

And also in Sect. 3.4., lines 340-341:

"*Since the W-band and Ka-band modules calibration is based on the G-band radar absolute calibration, the uncertainty in determining the W-band and Ka-band calibration factors primarily inherits the 1 dB error discussed in Sect. 3.2.*"

7. Figure 10, why do panels (a) and (d) have different amplitude scales? Should these not be the same, since we are comparing between the two?

   (Now Fig. 12) We do not intend to compare these two panels as the amplitudes shown are uncalibrated values that depend on the internal processing of each radar module. The calibrated reflectivities are shown in panels (c) and (f), which do show the same scale.

8. Figure 11 is great, and I feel that you could have a version of this figure earlier on (without the reflectivity ratios) or split this figure up, to illustrate to the reader the primary differences in the retrieved signals from each radar for the same cloud system. I was disappointed that this was left until the last figure, as it is excellent motivational material.

   We have added the following paragraph and Fig. 4 in Sect. 2.3., lines 156-162:

   "*An example of multifrequency reflectivities that can be found in the data provided with this article is plotted in Fig. 4. The combination of simultaneous observations at three greatly spaced frequency bands can reveal distinct cloud and precipitation features to further enhance the microphysical analysis. The process to obtain the calibrated data in Fig. 4, as well as dual-frequency ratios, and G-band Doppler spectra and moments, is described in Sect. 3.*"

[Figure]

*Figure 4: Example of CloudCube data on March 30, starting time 17:12:52 UTC, showing calibrated reflectivity at Ka (a), W (b) and G-band (c). The W-band plot (b) shows no data for approximately the first 500 m, corresponding to the blind range of the radar.*

9. Code availability. Is the processing code/QA code publicly available in some repository alongside the dataset? This is always nice to provide when possible.

The codes are not publicly available, but they can be made available upon request. We have added the following section and statement (lines 440-441):

**"5. Code availability**

*The processing codes can be made available upon request to the corresponding author."*

**Specific Comments:**

These are mostly small grammatical changes I would recommend for enhancing the overall readability of the manuscript.

**Line 38:** "this kind" -> "these kinds"

The recommendation has been accepted (line 38).

**Line 118:** during -> for

The recommendation has been accepted (line 123).

**Line 164:** What is ITU? I assumed it was a reference, but I don't see it in the reference list? Or was this defined somewhere else that I missed?

We have added the reference:

ITU-R P. 676-10 Recommendation: Attenuation by Atmospheric Gases, 2013. Available online: https://www.itu.int/dms_pubrec/itu-r/rec/p/R-REC-P.676-10-201309-S!!PDF-E.pdf, last access: March 29, 2024.

**Lines 234-236:** I would reorganize/rewrite this sentence, as it isn't clear to me what you mean here (I'd also delete "deep detail").

Please see next comment.

**Line 237:** I would also rewrite this sentence as "one can identify the times of most interest" is a bit verbose for what I think is being said.

We have rewritten and simplified this paragraph (lines 271-273):

"*The final G-band radar products consist of two separate data collections: one containing calibrated Doppler spectra (see Table 2 in Sect. 4), as in the example shown in Fig. 6c, and a second set with calibrated Doppler moments (see Table 3 in Sect. 4), as the ones presented in Fig. 7.*"

**Line 242:** "using for that purpose" -> using

The recommendation has been accepted (line 277).

**Line 287:** respond -> correspond

The recommendation has been accepted (line 322).

**Lines 335-336:** "can reveal valuable insight about the particle size and size distribution" -> "can reveal valuable insights into particle size distributions"

The recommendation has been accepted (lines 372-373).

**Line 336:** performing -> deriving

The recommendation has been accepted (line 373).

---

## Author Response (AR2)

Dear Referee,

Thanks for pointing out this issue that we overlooked when reprocessing the data. We have fixed it in a new version of the data files. Please check https://doi.org/10.5281/zenodo.10076227.

Sincerely,
The Authors

**Revision comment**

The author's substantial improvements to dataset readability and accessibility are greatly appreciated. The additional file metadata and clarity around missing values will aid in streamlining the use of this data in future work. Further, the authors have satisfactorily answered my general science questions and fixed the previously listed technical errors. I believe the manuscript is much improved overall, and now tells a very clear narrative surrounding the utility of the G-band radar.

My only comment before fully recommending the paper for publication resides in the revision to the data itself. During editing, it appears as though a time step issue has arisen (perhaps introduced during compression/file deflation?) where time axis values (i.e., time and time_offset) are no longer sets of unique values. I've included an example below for the first 10 time entries in 20230417_200402_Gband_Spectra.nc (however, this appears to be the case for all the cases I looked at):

['2023-04-17T20:04:01.999998208'        '2023-04-17T20:04:01.999998208'
'2023-04-17T20:04:01.999998208'         '2023-04-17T20:04:03.000002816'
'2023-04-17T20:04:03.000002816'         '2023-04-17T20:04:03.999997696'
'2023-04-17T20:04:03.999997696'         '2023-04-17T20:04:03.999997696'
'2023-04-17T20:04:05.000002048'         '2023-04-17T20:04:05.000002048']

This is likely an easy fix, but an important one, as currently it is impossible to tell which of two values comes first in the series. To see this error clearly, you can try opening one of these datasets in a common NetCDF visualization program like NASA Panoply (it will give you an error).

With this issue fixed, I believe the manuscript and dataset will be ready for publication in ESSD.